# Karyotypic Flexibility of the Complex Cancer Genome and the Role of Polyploidization in Maintenance of Structural Integrity of Cancer Chromosomes

**DOI:** 10.3390/cancers12030591

**Published:** 2020-03-05

**Authors:** Christina Raftopoulou, Fani-Marlen Roumelioti, Eleni Dragona, Stefanie Gimelli, Frédérique Sloan-Béna, Vasilis Gorgoulis, Stylianos E. Antonarakis, Sarantis Gagos

**Affiliations:** 1Laboratory of Genetics, Center of Experimental Medicine and Translational Research, Biomedical Research Foundation of the Academy of Athens, 11527 Athens, Greece (BRFAA); xraftopoulou@yahoo.gr (C.R.); fmroumelioti@gmail.com (F.-M.R.); edragona@bioacademy.gr (E.D.); 2Department of Genetic Medicine and Development, University of Geneva Medical School, 1211 Geneva, Switzerland; stefania.gimelli@hcuge.ch (S.G.); frederique.bena@hcuge.ch (F.S.-B.); Stylianos.Antonarakis@unige.ch (S.E.A.); 3Histology-Embryology Laboratory, Medical School, National Kapodistrian University of Athens, 11517 Athens, Greece; vgorg@med.uoa.gr

**Keywords:** chromosomal instability in neoplasia (CIN), alternative lengthening of telomeres (ALT), whole genome doubling (WGD), polyploidy, DNA replication stress, intratumor genomic heterogeneity, therapy resistance

## Abstract

Ongoing chromosomal instability in neoplasia (CIN) generates intratumor genomic heterogeneity and limits the efficiency of oncotherapeutics. Neoplastic human cells utilizing the alternative lengthening of telomeres (ALT)-pathway, display extensive structural and numerical CIN. To unravel patterns of genome evolution driven by oncogene-replication stress, telomere dysfunction, or genotoxic therapeutic interventions, we examined by comparative genomic hybridization five karyotypically-diverse outcomes of the ALT osteosarcoma cell line U2-OS. These results demonstrate a high tendency of the complex cancer genome to perpetuate specific genomic imbalances despite the karyotypic evolution, indicating an ongoing process of genome dosage maintenance. Molecular karyotyping in four ALT human cell lines showed that mitotic cells with low levels of random structural CIN display frequent evidence of whole genome doubling (WGD), suggesting that WGD may protect clonal chromosome aberrations from hypermutation. We tested this longstanding hypothesis in ALT cells exposed to gamma irradiation or to inducible DNA replication stress under overexpression of p21. Single-cell cytogenomic analyses revealed that although polyploidization promotes genomic heterogeneity, it also protects the complex cancer genome and hence confers genotoxic therapy resistance by generating identical extra copies of driver chromosomal aberrations, which can be spared in the process of tumor evolution if they undergo unstable or unfit rearrangements.

## 1. Introduction

From the pivotal work of Peter Nowell, tumor cell populations are known to undergo continuous genome evolution following Darwinian processes of selection and adaptation [1]. These biological processes are driven by genomic instability that accompanies neoplastic cell growth from the initial steps of carcinogenesis up to the last stages of metastases [2].

Cancer genomes may acquire all types of known mutations, from single nucleotide substitutions up to extensive copy number alterations of large chromosome segments, whole chromosomes, or even sets of chromosomes [3]. Telomere dysfunction, oncogene-induced replication stress, and deficient DNA repair responses contribute to the genomic instability in cancer [4,5]. Chromosomal instability in neoplasia (CIN) has long been recognized as the most common form of genomic instability in cancer [6]. CIN is present in all types and stages of human cancer, substantially affecting the dosage, integrity, and architecture of the genome via structural or numerical chromosomal aberrations [7,8]. Chromosomal imbalances emerge at the early steps of carcinogenesis [9], are often randomly distributed among the cancer cells [8], activate oncogenic pathways, and inactivate tumor suppressors [10]. Furthermore, continuous CIN generates widespread intratumor genomic heterogeneity, allowing selective processes that drive cancer cell populations into malignancy, limiting in parallel the efficiency of oncotherapeutic schemes [11,12]. A better understanding of CIN is critical not only for deciphering fundamental aspects of carcinogenesis but also for achieving efficient therapies against advanced malignancies [3,8,13].

Most cancer genomes evolve via polyploidy/heteroploidy and display widespread aneuploidy [14,15,16,17,18,19]. Although there are diverse roots to polyploidy [20], polyploidization in cancer-genome evolution appears to occur mostly via whole genome doubling (WGD), through the process of endomitosis, and is more frequent in advanced malignancies [17,19,21]. Endomitosis, or endocycling, generates polyploid nuclei with clustered centrosomes that retain the abilities to divide [20]. Polyploidization via WGD in cancer cells has been associated with increased rates of numerical chromosome instability [22,23]. Importantly, increased polyploidization frequencies are associated with resistance to radiotherapy and several types of genotoxic oncotherapy [23,24,25,26]. Hence, polyploidy has been proposed as a buffering process that rescues cancer cells from deleterious mutations acquired during carcinogenesis [22,27]. However, although polyploid tumor cells were thought to adapt and evolve more efficiently than their near-diploid counterparts [28], in microsatellite unstable but chromosomally stable cancer cells, they rarely persist their progenitor clones, and hence they may be negatively selected [22]. Nevertheless, a form of massive ploidy increase, followed by de-polyploidization (or polyploidy reduction) termed “neosis”, was proposed to confer resistance to genotoxic oncotherapy [29,30,31].

Telomeres are repetitive nucleoprotein chromosomal organelles that protect the ends of linear DNA molecules from being processed as sites of DNA damage [4,32]. Telomeres are progressively depleted after each round of DNA replication in most dividing human somatic cells [4]. Extreme telomere shortening activates the DNA-damage-repair machinery aggravating terminal chromosome fusions and structural CIN [33]. Telomere driven genomic instability is accompanied by chromosomal breakage–fusion–bridge cycles (B/F/B) and polyploidization [16,17,18,19,34,35,36], and it produces extensive aneuploidy through anaphase lagging or micronuclei formation [37,38].

In normal somatic cells, telomere restoration is suppressed, while senescence or apoptosis prevent neoplastic transformation [4,5]. To bypass these barriers, approximately 85–90% of human malignancies activate the reverse transcriptase telomerase [39]. About 10–15% of human tumors utilize the alternative lengthening of telomeres (ALT) [40]. ALT is a complex, telomerase-independent pathway, driven by homology-mediated DNA-replication repair [41]. Cancer-derived or immortalized human cell lines exclusively utilizing the ALT pathway of telomere maintenance exert 4–5 times more pronounced rates of random structural chromosomal instability as compared to telomerase positive cells [36]. In addition, ALT cells frequently undergo WGD due to increased endogenous telomere dysfunction [19]. Therefore, the ALT pathway provides an ideal context to study the contribution of CIN and WGD in tumor evolution and therapy resistance.

We applied molecular cytogenetic assays and array-based whole genome dosage analyses in a panel of continuous human ALT cell lines exposed to DNA replication stress, gamma irradiation, or chronic chemotherapy-induced genotoxic damage in order to explore the extent of the karyotypic flexibility of the ALT cancer genome and to unravel patterns of neoplastic genome evolution. In addition, we tested, for the first time in chromosomally unstable cells, the hypothesis that polyploidization contributes to buffering of the cancer genome from lethal hypermutation [8,24,42]. Our data reveal a remarkable tendency of the cancer cells to sustain en masse, specific genomic imbalances, despite the continuous karyotypic alterations introduced by double-strand breaks and erratic inter-chromosome ligations. In this context, polyploidization followed by a variable amount of chromosome losses, protects the structural integrity of the abnormal cancer genome from ongoing chromosomal mutations, while in parallel, facilitates the mitotic tolerance of novel clonal rearrangements that increase intratumor genomic heterogeneity, paving the way for the acquisition of genotoxic oncotherapy resistance.

## 2. Results

### 2.1. Clonal Evolution of the ALT Cancer Genome Is Characterized by Narrow Karyotypic Flexibility and the Tendency to Maintain Cancer-Genome Dosage Stability

To address the flexibility of the complex ALT genome, and to unravel patterns of cancer genome evolution driven by oncogene induced replication stress, telomere dysfunction, or genotoxic damage, we examined five isogenic cell lines of the human ALT osteosarcoma cell line U2-OS, representing different karyotypic outcomes, using M-FISH/ (Multi-Color Fluorescent In Situ Hybridization) molecular karyotyping, combined with inverted DAPI banding and aCGH (array-Comparative Genomic Hybridization). Our U2-OS cell line panel included two independently obtained unperturbed (wild-type) cells (wt1 and wt2) [43], two isogenic chemo-resistant wt2-derived cell lines (R1 and R2) [44], and another wt2-derivative cell line that stably overexpresses the replication licensing factor CDT1 [45]. The five U2-OS cell lines displayed similar numerical chromosomal constitution, with predominant peri-triploid subclones composed of 69–80 chromosomes. In addition, they all exerted similar heteroploidy rates, with co-dividing, endoreduplicated populations of 115–136 chromosomes, ranging from 10% to 20% of the metaphase spreads. Based on the presence or absence of specific clonal structural chromosome rearrangements, we identified at least two karyotypically distinct, major subclones per independent U2-OS-derivative cell line (Figure 1A and Appendix A). Although diverse, all 10 representative U2-OS karyotypes shared at least six identical structural chromosome rearrangements, confirming monoclonal origin from a single progenitor genome. At least four events of the U2-OS chromosomal evolution coincided with karyotypes bearing multiple duplicated copies of clonal recombinant chromosomes, suggesting that polyploidization and polyploidy reduction are frequent events during ALT tumor clonal evolution in culture. The U2-OS karyotype was shaped by whole or partial chromosome gains or losses and by novel clonal structural rearrangements such as unbalanced translocations, deletions, or multicentric chromosome formations. While clonal evidence of chromoanagenesis [46] was rare by M-FISH, specific recombinant chromosomes or segments displayed a tendency to persist in the evolving karyotype, even after breakage and rejoining into novel marker chromosomes (Figure 1A–C). Since these jumping recombinant DNA segments were present in all karyotypically different U2-OS cell lines, we presumed that they may confer critical genome imbalances supporting continuous cellular growth. We tested this hypothesis by aCGH. Despite the karyotypic diversity, all five U2-OS cell lines shared a plethora of large identical duplications/deletions (Figure 1D and Appendix A). These observations reveal a limited karyotypic flexibility of the highly unstable ALT genome, suggesting a trend of the dividing cancer cells to maintain specific dosage imbalances, despite the elevated recombination rates generated by increased breakage and illegitimate chromosome rejoining.

### 2.2. Distribution of Random Structural Chromosome Anomalies between Co-Dividing ALT Cells and a Putative Role of Polyploidy in the Protection of Cancer Genome Integrity

Virtually all chromosomes of the human ALT karyotype can be involved in clonal or random structural rearrangements [35,36]. In this study, we sought to investigate the distribution of random structural chromosome rearrangements per metaphase in a population of co-dividing mitotic cells from a panel of four human ALT cell lines. The cell lines examined were Saos-2 (osteosarcoma), LS-2 (liposarcoma), and the immortalized VA-13 and GM-847 [47,48]. Molecular karyotyping combined to inverted DAPI banding, was performed in 15 co-dividing metaphase spreads/cell lines, randomly selected based on staining by high quality multicolor-FISH. Random (non-clonal) structural rearrangements were considered as all cytogenetically visible aberrations that were identified only once in a population of 15 co-dividing cells. Clonal events were considered as all rearrangements that were present in two or more mitotic nuclei or found in duplicated/multiplicated copies per karyogram, implying at least two cell divisions. Clonal rearrangements were not included in CIN counts. Remarkably, all four cell lines displayed a non-uniform distribution of structural CIN between mitotic cells (Figure 2A and Appendix A). Many cells displayed high incidence of novel structural rearrangements, whereas others appeared protected from random recombinatorial events. To identify karyotypic determinants of genomic stability, we examined the molecular karyotypes of the cells presenting the lower rates of random structural rearrangements. Strikingly, 50% of these cells were byproducts of whole genome endoreduplication, followed by various degrees of chromosome losses (Figure 2B and Appendix A). These results suggest that the effects of endogenous genotoxic stressors are not evenly dispersed among co-dividing immortalized ALT cells. In this context, polyploidization followed by chromosome losses (polyploidy reduction) appears to protect cancer genome integrity from randomly emerging structural chromosomal instability.

### 2.3. Polyploidization Protects the Abnormal Cancer Genome from Extreme Structural CIN and Promotes Intratumor Genomic Heterogeneity

It has been suggested that polyploidy combined with increased rates of chromosome losses may buffer deleterious mutations produced by the hypermutator cancer phenotype [8,24,42]. To test this hypothesis at the single-cell level, we exposed the human immortalized ALT cell line VA-13 to 2.4 Gray of gamma irradiation. Molecular karyotyping was performed to examine the frequency and distribution of random structural aberrations between representatives of the major cell populations and cells from the same culture presenting endoreduplicated genomes. As anticipated, 10 d post irradiation, 15 mitotic cells with the most representative (modal) chromosome number of 64–78 chromosomes displayed a significant increase in random structural rearrangements, as compared to 15 non-irradiated VA-13 cells (*p* = 0.03 by One-way ANOVA). We then examined 30 irradiated metaphase spreads from the same harvest, divided into two, randomly picked groups, based on numerical chromosomal constitution. Interestingly, the cell group composed by 15 irradiated endoreduplicated metaphase spreads of 104–178 chromosomes showed significantly lower rates of random structural chromosome anomalies as compared to co-dividing cells of the modal chromosome number (61–55 chromosomes) (*p* < 0.0001 by One-way ANOVA), (Figure 3A and Appendix A). Conditional chronic overexpression of the Cyclin Dependent Kinase Inhibitor 1A (CDKN1A-known as p21) in the human ALT osteosarcoma cell line Saos-2 deregulates replication licensing and generates increased rates of structural chromosome instability [49]. We analyzed by M-FISH and inverted DAPI banding twenty randomly picked, Saos2 p21WAF1/Cip1 Tet-ON cells, survivors of the p21-induced replicative crisis, and twenty isogenic control cells, separated into two groups: those representing the major hypotriploid clones and those showing an endoreduplicated genome. Again, the rates of structural instability were lower in the endoreduplicated cells as compared to co-dividing cells with 51–56 chromosomes belonging to the prevalent clones (Figure 3B, Appendix A) (*p* = 0.013 by One-way ANOVA). These results support a protective role of polyploidization over replication-stress-driven structural CIN. Chromosome counts in 50 randomly picked VA-13 metaphase spreads from four subsequent passages upon gamma irradiation (2–4 days per passage) showed a gradual decrease in the frequencies of WGD. Chromosome numbers of endoreduplicated nuclei varied from 83 to 165 chromosomes. Most WGD cells displayed chromosome numbers between 100–119, but they rarely exceeded 120. In fact, cells with more than 120 chromosomes were derived from more than one WGD rounds, as verified by multiple numbers of identical copies of marker chromosomes (Appendix A). With very few exceptions, we did not observe dividing cells with gradual chromosome losses and intermediate chromosome numbers between the distinct ploidy levels, suggesting that cells undergoing a high extent of chromosome losses are not able to divide (Figure 3C and Appendix A). Notably, a marked increase in karyotypic heterogeneity and intra-specimen subclonality was evident in polyploid cells that survived gamma irradiation or p21-induced replicative crisis (Figure 3D, Appendix A). In both metaphase groups, several of the endoreduplicated genomes displayed in duplicate copies, novel clonal structural aberrations not observed in the non-endoreduplicated, co-dividing cells. Collectively, our results suggest that endoreduplication generates extra identical copies of driver chromosomal aberrations that can be spared in the process of evolution if they undergo unstable or unfit rearrangements. Multiple stochastic chromosome losses and novel structural aberrations appear to be mitotically tolerated by cancer cells undergoing WGD. Hence, in chromosomally unstable cancers, endoreduplication perpetuates the integrity of driver chromosome aberrations and facilitates the generation of intratumor genomic heterogeneity.

## 3. Discussion

Most mammalian cancers derive from a single progenitor cell that acquires the potential for uncontrolled continuous proliferation [3,10]. Cancer is associated with impaired DNA damage responses and insufficient cell cycle check-point controls that predispose cells to genomic instability [9]. Eventually, the progeny of any cancer-originating cell will be exposed to further oncogene-induced DNA replication stress [5], telomere dysfunction [50], and oxidative or metabolic stressors [51]. These insults produce DNA double-strand breakage and illegitimate chromosome fusions that continuously jeopardize the integrity of the cancer genome.

During carcinogenesis, cancer cells stochastically accumulate a combination of genomic oncogenic aberrations that sufficiently sustain continuous malignant growth. Such driver mutations are positively selected among tumor cell populations [3]. However, according to the mutator phenotype hypothesis, every part of the cancer genome is prone to deleterious mutations that are equally capable of affecting normal and mutated cancer-promoting genes or genomic imbalances [42,52]. Hence, cancer genome evolution can be considered as a continuous equilibrium between the preservation of specific tumor promoting mutations and their losses. In a stochastic course of genome-altering events, cancer cells may benefit from the emergence of novel mutations that will be clonally preserved if they allow continuous cellular growth or confer novel selective advantages. Novel mutations that reduce cancer genome fitness are expected to be negatively selected from the ongoing tumor population.

Despite the increased rates of endogenous and exogenous CIN in the snapshot of clonal evolution of the U2-OS cell line recorded in this study, we encountered limited karyotypic flexibility of the immortalized osteosarcoma genome. Representative chromosome numbers and heteroploidy rates were similar between diverse evolutionary outcomes. Many chromosome alterations were identically shared between the different major U2-OS side lines, and most importantly, the aCGH comparison revealed a striking propensity of the cancer genome to sustain specific gross genomic imbalances regardless of widespread structural karyotypic changes. These findings are in line with the hypothesis that the immortalized genome tends to sustain those specific genomic characters that mostly represent the progenitor cancer cell, so to better adapt and multiply into the culture microenvironment [53].

The chromosomally aberrant cancer-derived or transformed continuous human cell lines utilized in laboratory research are monoclonal in origin and can be identified by their unique numerical and structural karyotypic constitution [54,55,56]. This karyotypic signature can authenticate a specific cell line because it is maintained through subsequent passages and cycles of freezing and thawing, despite diverse culture conditions and various degrees of endogenous ongoing CIN [57]. However, genetic manipulations or genotoxic stress may alter the relatively stable karyotypic constitution of immortalized cell lines [58]. As described in our study, the karyotypic evolution of the osteosarcoma cell line U2-OS under chronic doxorubicin exposure or CDT1 overexpression suggests that despite ongoing double-strand breaks (DSBs) and illegitimate recombination, the extent of novel chromosomal changes that can be mitotically tolerated in immortalized cells in culture is not unlimited. Our results indicate that novel alterations may reshuffle the cancer genome, but they must also retain, as undisturbed as possible, the ongoing equilibrium of losses and gains of genomic materials that are compatible with continuous cellular growth.

The evolutionary persistence of specific chromosome aberrations through jumping translocations indicates their importance for continuous cell growth. Jumping translocations are frequently encountered in cytogenetic analyses of solid tumor cell lines [58,59]; they have been reported in hematological malignancies [60] and in the genome evolution of prostate cancer [61,62]. Chromothripsis occurs frequently in cancer cells suffering from DNA replication stress or telomere dysfunction [63,64]. Chromoanagenesis, the clonal outcome of chromothripsis, implies reconstitution of one or more hyperfragmented chromosomes into a new recombinant DNA molecule [46]. In this study, chromoanagenesis was microscopically evident in the evolution of the doxorubicin-resistant R2 cells as a single clonal event that produced a novel recombinant chromosome. This is consistent with the notion that most large-scale chromothripsis events may lead to chromosome losses [63] and hence they may be excluded from tumor genome evolution.

Whereas the evolution of the major dividing subclones of the U2-OS cell line was mainly shaped by chromosome losses and novel structural rearrangements, at least four evolutionary events coincided with karyotypes bearing duplicate copies of abnormal or apparently intact chromosomes. A possible explanation for these findings is that these subclones originated from WGD followed by massive chromosome losses. The rates of polyploidy in tumor specimens and cancer cell lines are found to be increased after exposure to ionizing radiation or genotoxic agents, implying that polyploidization may act as a DNA-damage response [65].

Previous studies support the hypothesis that polyploidization accelerates numerical chromosomal instability (CIN) and in this way leads to aneuploidy and the karyotypic evolution of a transformed phenotype [22,23,66]. When we examined the distribution of random structural chromosome rearrangements between co-dividing mitotic cells in a panel of four human ALT cell lines, we found that 50% of the metaphases exhibiting a relatively low random structural chromosome aberration load displayed evidence of WGD. Hence, endoreduplication followed by variable degrees of chromosome losses may protect the aberrant cancer genome from ongoing, handicapping chromosome aberrations. We tested this assumption in ALT cells exposed to gamma irradiation or p21-induced replication stress. Molecular cytogenetic analyses revealed that although elevated, as compared to wild type or control cells, the rates of random structural CIN were significantly suppressed in the groups of endoreduplicated co-dividing cells. Hence, cells undergoing WGD can better tolerate ongoing structural CIN, probably because they can afford to lose a limited number of multicentric or unbalancing chromosomal rearrangements without major consequences in their dividing capacity. Dewhurst et al. [22], showed that tetraploid progeny had a greater tolerance of chromosome segregation errors relative to diploids. Nevertheless, massive chromosome losses may kill the cancer cell or disrupt its proficiency to divide. In VA-13 metaphase spreads examined in subsequent passages after gamma irradiation, we observed a gradual decrease in the original high frequencies of hyperploidy during continuous cell growth, without a concomitant increase in intermediate chromosome numbers spanning the discrete ploidy indices. This suggests that newly endoreduplicated cells can afford to lose only a limited number of chromosomes and continue to divide. Massive polyploidy reduction through multiple chromosome losses produces extreme genomic imbalances that jeopardize mitotic efficiency. Nevertheless, if a mitotically handicapped polyploid cell survives to lose almost half of its endoreduplicated genome, so as to reach the combination of imbalances of the major aneuploid clones, it may again become mitotically active and capable of perpetuating immortal growth. This is consistent with the “polyploidy remnants” in the karyotypic evolution of U2-OS or VA-13 cells recorded in this study and may propose a genomic status for dormant cancer cells [67].

Subclonal karyotypic heterogeneity was dramatically elevated in the groups of endoreduplicated cells surviving severe genotoxic stressors. In addition, novel endoreduplicated clones displayed multiple newly formed complex chromosomal rearrangements in duplicate or multiple copies. These rearrangements were not present in the most prevalent, non-endoreduplicated subclones, suggesting a mitotic handicap that was counteracted by WGD. Hence, novel structural chromosomal aberrations appear to be better tolerated in the reduplicated context, which allows their mitotic perpetuation to increase intratumor genomic heterogeneity.

A large body of experimental and clinical evidence suggests that the frequencies of WGD are highly pronounced in cancer cells surviving genotoxic therapeutic interventions and may act as a mechanism of therapy resistance [23,24,25,26,29,68,69]. In line with these findings, our cytogenetic results uncover a key role of polyploidization in the protection of the integrity of complex cancer genomes by ameliorating the effects of large scale endogenous or exogenous genotoxic insults.

Compared to next generation sequencing (NGS), molecular cytogenetics is limited in resolution and throughput capacity, but can still provide important information on ploidy and large scale randomly occurring chromosome mutator events at the single-cell level [70]. Moreover, M-FISH examines hundreds of chromosomes in mitotic cells that evidently retain a tumor-propagating ability. A significant proportion of cancer cells undergo cell death or remain mitotically arrested [71]. The genomic content of such cells may not be representative of genomic evolution.

The endogenous, highly increased rates of CIN operating in the ALT pathway provide a unique biological context to study karyotypic evolution during continuous cell growth in a fast forward mode [36]. Interestingly, intratumor heterogeneity and karyotypic evolution appear stochastically oriented, but their outcome is deterministic. Hence, along with oncogenic sub-microscopic driver mutations that were not examined in this study, our results show for the first time that WGD followed by chromosome losses preserves cancer genome integrity and facilitates the maintenance of a pre-established equilibrium of gross genomic imbalances that promote continuous growth. A better understanding of the biological mechanisms leading to polyploidy will provide novel tools to cure advanced cancers.

## 4. Materials and Methods 

### 4.1. Cell Lines and Culture Conditions

The ALT osteosarcoma cell lines Saos-2 and U2-OS wt1 were obtained from the European Type Culture Collection (Wesel, Germany). The osteosarcoma cell line U2-OS wt2 was a gift from E. Gonos (National Hellenic Research Foundation, Athens, Greece). The SV-40 large T-antigen transformed ALT cell lines GM-847 and VA-13 were donated by A. Londoño-Vallejo (Institute Curie, Paris, France) [47]. The ALT liposarcoma LS-2 cells were kindly provided by D. Broccoli (Fox Chase Cancer Center, Philadelphia, PA, USA) [48]. The CDT1 overexpressing U2-OS and Saos2 p21WAF1/Cip1 Tet-ON cells were produced in the Gorgoulis lab and are described in [45,49]. The U2-OS CDT1 Tet-ON cells were grown in the presence of 0.5 µg/ml Doxycycline (Sigma-Aldrich, St. Louis, MO, USA) for 55 days before harvest. The U2-OS wt2 derivative R1 and R2 Doxorubicin-resistant cell lines, are described in [44]. All cell cultures were grown at 37 °C and 5% CO2 in Dulbecco’s modified Eagle’s medium (Gibco, Grand Island, NY, USA) supplemented with 10% FBS (Gibco, Life Technologies Ltd, Paisley, UK), 25 units/mL penicillin (Sigma, St Louis, MO, USA), and 25 pg/mL streptomycin (Invitrogen, Grand Island, NY, USA). Before conventional cytogenetic harvest, the U2-OS R1 and U2-OS R2 cells were grown for a week in the presence of 0.0035 μM or 0.035 μM of Doxorubicin, Hydrochloride (Calbiochem, San Diego, CA, USA).

### 4.2. Cytogenetics

Logarithmically growing cell cultures were exposed to Colcemid (0.1 µg/mL) (Gibco) for 1 to 3 h, at 37 °C in 5% CO2. Cells were harvested by trypsinization (Gibco), suspended in medium, and spun down (10 min at 1000 rpm). Supernatant was removed completely, and 5 mL of 0.075 M KCl (Sigma) at room temperature was added drop-by-drop. The cells were incubated for 20 minutes at room temperature, and then 1 ml of fixative [3× methanol (Applichem GmbH, Darmstadt, Germany)/1× CH3COOH (Merck, Darmstadt, Germany)] was added. Cells were spun down (10 min at 1000 rpm), supernatant was removed, fixative was added, and the cells were re-centrifuged for 10 min at 1000 rpm. Finally, cells were dropped onto wet microscope slides and left to air-dry. For karyotypic analysis, we combined inverted DAPI staining and molecular karyotyping by multicolor-FISH (MetaSystems GmbH, Altlussheim, Germany). M-FISH was performed according to the manufacturer’s protocols (MetaSystems). For inverted DAPI banding, slides were counterstained and mounted with 0.1 µg/mL DAPI in VECTASHIELD antifade medium (Vector Laboratories, Burlingame, CA, USA). Cytogenetic analyses were performed using a ×63 magnification lens on a fluorescent Axio-Imager Z1, Zeiss microscope (Zeiss, Oberkochen, Germany), equipped with a MetaSystems charge-coupled device camera and the MetaSystems Isis software (version Isis 5.1.22, MetaSystems GmbH, Altlussheim, Germany).

### 4.3. Comparative Genomic Hybridization

We obtained genomic DNA from normal male 46,XY, from Promega (Promega, Madison, WI, USA). Genomic DNA was prepared from each cell line by using the QIAamp DNA Blood Mini kit (Qiagen). For each CGH hybridization, we digested 1 μg of genomic DNA from the reference (46,XY male) and the corresponding experimental sample with AluI (10 U/μL) and RsaI (10 U/μL) (Promega) at a final volume of 20.2 μL. All digests were done for 2 h at 37 °C and then verified by analysis of 1% agarose gel. Labelling reactions were performed with Agilent Genomic DNA Labelling Kit PLUS. Random primers were added to the digested DNA, incubated at 95 °C for 3 min, and then moved to ice and incubated for 5 min. According to the manufacturer’s instructions, the labelling master mix was prepared on ice in a final volume per reaction of 21 μL, containing Nuclease-free water 2 μL, 5× Buffer 10 μL, 10× dNTP 5 μL, Cyanine 3-dUTP (1 mM) for the 46,XY male reference or Cyanine 5-dUTP (1mM) for the experimental sample 3 μL, and Exo-Klenow fragment 1 μL. Labelling master mix was added in each tube, reaching a final volume of 50 μL. They were incubated for 2 h at 37 °C, and then at 65 °C for 10 min to inactivate the enzyme. Experimental and reference labelled targets were combined and mixed with 430 μL TE 1× at pH 8. Using a Micron centrifugal filter device kit (Millipore) we transferred the samples in microcorn tubes and they were centrifuged for 10 min at 8000 rpm. Subsequently, we discarded the supernatant and added 480 μL of TE 1× pH8 and centrifuged the samples again for 10 min at 8000 rpm. The cyanine 5- and cyanine 3-labelled gDNA mixtures, after being purified, concentrated with a Centricon YM-30 column, and resuspended to a final volume of 41 μL, were then mixed with 5 μL of human Cot-1 DNA (1 mg/mL) (Invitrogen), 11 μL Agilent 10× Blocking agent, and 55 μL Agilent 2× hybridization buffer according to the Agilent Oligo aCGH Hybridization Kit (Agilent, Santa Clara, CA, USA). Before hybridization to the array, the 110 μL hybridization mixtures were denatured at 95 °C for 3 min and incubated at 37 °C for 30 min. Then, 100 μL of the hybridization mixture was applied to the 4× 44K array by using an Agilent microarray hybridization chamber (Agilent, Santa Clara, CA, USA). Hybridization was carried out for 40 h at 65 °C in a rotating oven at 20 rpm. The arrays were then disassembled in Oligo aCGH Wash Buffer 1, washed by stirring in Oligo aCGH Wash Buffer 1 for 5 min, then washed with Oligo aCGH Wash Buffer 2 for 1 min at 37 °C, followed by 1 min at room temperature in 100% acetonitrile. Slides were dried and scanned by using an Agilent DNA microarray scanner. Microarray images were analyzed by using the Feature Extraction software (version 10.5.1.1, Agilent Technologies, Santa Clara, CA, USA). The Human Genome CGH Microarrays 44K represent 43,000 coding and noncoding human sequences, the well characterized genes are represented by 1 probe and the cancer relevant genes by 2 probes, and they have a spatial resolution of 43 Kb overall median probe.

### 4.4. γ Irradiation

The VA13 cell line grown as sub-confluent monolayers in T-75 flasks was irradiated with 2.4 Gy of γ-rays in a γ-cell 220 irradiator (Atomic Energy of Canada Limited, Ottawa, Canada) with a dose rate of 0.5 Gy/min. Cells were incubated for 24 h and subcultured. Metaphases from parallel cultures were collected in different time intervals.

### 4.5. Statistical Analyses

All statistical analyses were carried out using the MiniTab software (version 14, MiniTab, State College, PA, USA).

## 5. Conclusions

The complex genome of neoplastic cells utilizing the alternative lengthening of telomeres (ALT)-pathway, despite extensive karyotypic evolution, maintains a relatively stable set of large imbalances compatible with continuous proliferation. In the ALT context that continuously generates DNA double-strand breaks (DSB) due to oncogene-induced replication stress and telomere dysfunction, the faithful mitotic perpetuation of clonal structural chromosomal aberrations, which are the carriers of oncogenic mutations and critical imbalances, is imperative for cancerous growth. In cells belonging to the prevalent highly aneuploid sub-populations, erratic DNA repair of endogenous or therapy-induced genotoxic insults may lead to chromosome losses that jeopardize the equilibrium of tumor propagating genomic anomalies and the ability to divide.

Whole genome doubling via endoreduplication, already known as a response to DNA damage and as a means of tolerance of chromosome losses or novel structural aberrations in the cancer genome, generates identical extra copies of clonal recombinant chromosomes, which are able to be lost if they are involved in random, unfit, or unbalancing aberrations. This ongoing selective process of whole chromosome-xeroxing and losses, preserves the integrity of driver oncogenic mutations carried by clonal recombinant chromosomes, sustaining the genetic information that dictates neoplastic growth and may confer genotoxic therapy resistance. A better understanding of polyploidization is critical not only to deciphering fundamental aspects of carcinogenesis but also for achieving efficient therapies against advanced malignancies.

## Figures and Tables

**Figure 1 cancers-12-00591-f001:**
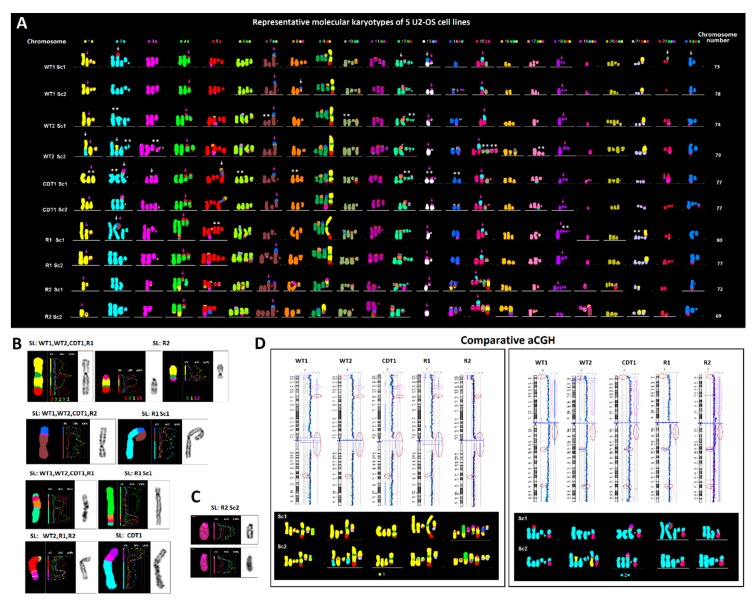
Karyotypic evolution in the U2-OS osteosarcoma cell line. (**A**) Representative, pseudo-colored, multicolor-FISH karyotypes of ten major subclones (Sc), from five U2-OS-derivative cell lines (SL) composed of 69–80 chromosomes. In U2-OS cells, every homologue of the human karyotype is affected by clonal structural or numerical aberrations. Despite the karyotypic diversity, a monoclonal origin of all side lines is evident by the common presence of six identical recombinant chromosomes (pink arrows). Based on the constitution of chromosome 5, the WT1 cells probably represent the most ancestral population. The CDT1-overexpressing, and Doxorubicin-resistant R1 and R2 cells derive from WT2. White arrows depict clone-specific rearrangements. Note that several evolutionary steps (i.e., the process from WT1 to WT2 Sc1-2 or the evolution of CDT1 and R1 cells) were accompanied by karyotypes bearing multiple duplicated copies of identical clonal recombinant chromosomes (asterisks) suggesting that leaps in karyotypic evolution were accompanied by whole genome duplication (WGD) followed by multiple chromosome losses. (**B**) Jumping translocations of large recombinant segments clonally present in most U2-OS-derivative cell lines and subclones. (**C**) Chromoanagenesis was recorded only in one recombinant chromosome, composed of multiple alternate segments of chromosomes 5 and 19, in the chemo-resistant R2 cells. (**D**) Identical large genome imbalances identified by aCGH are stably maintained between the U2-OS-derivative cell lines despite the karyotypic alterations produced by extensive chromosome breakage and illegitimate rejoining. Red circles indicate the presence of the same imbalance in all 5 cell lines, pink circles depict undistinguishable duplications/deletions in 4 out of 5 karyotypically-diverse U2-OS cell lines. Lower boxes include partial karyograms involving genomic material of the chromosome analyzed by aCGH, representing two major subclones per side line.

**Figure 2 cancers-12-00591-f002:**
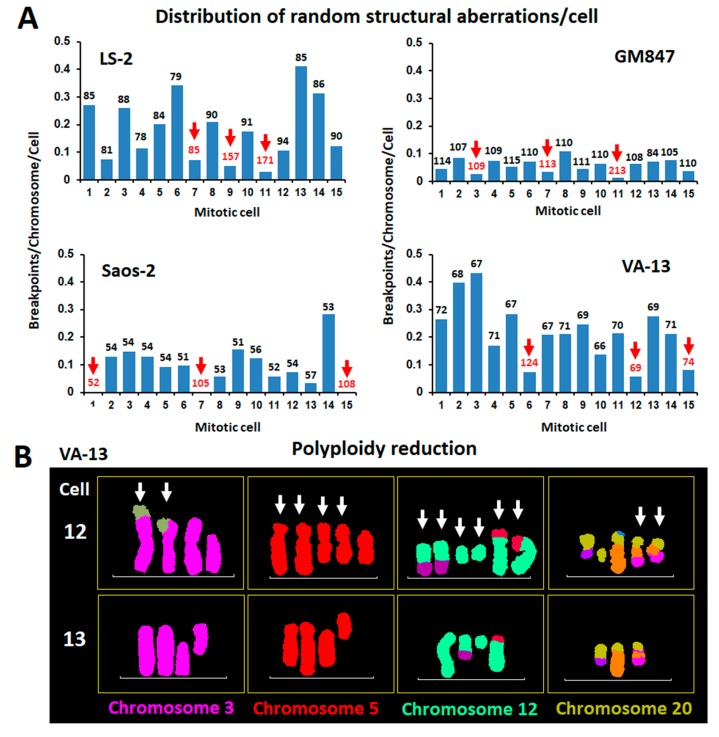
Distribution of random structural chromosomal instability in neoplasia (CIN) among co-dividing alternative lengthening of telomeres (ALT) cells and the possible role of WGD in the perpetuation of the integrity of driver chromosomal rearrangements. (**A**) Uneven distribution of non-clonal structural chromosomal rearrangements between 15 randomly picked, co-dividing cells from 4 human ALT cell lines. Bars indicate structural CIN load, calculated as breakpoints/chromosome. Chromosome number/metaphase is indicated above each data bar. From every cell line we selected 3 cells with the lower rates of structural chromosomal instability. Interestingly, half of the 12 CIN escapee cells (red arrows and numbers) were byproducts of WGD or showed evidence of WGD reduction. (**B**) Pseudo-colored partial karyograms of two co-dividing VA-13 mitotic nuclei belonging to the major clones and each bearing 69 chromosomes. In contrast to #13, cell #12 displays extremely low rates of structural CIN and presents several duplicated copies of clonal recombinant chromosomes, suggesting WGD followed by multiple chromosome losses.

**Figure 3 cancers-12-00591-f003:**
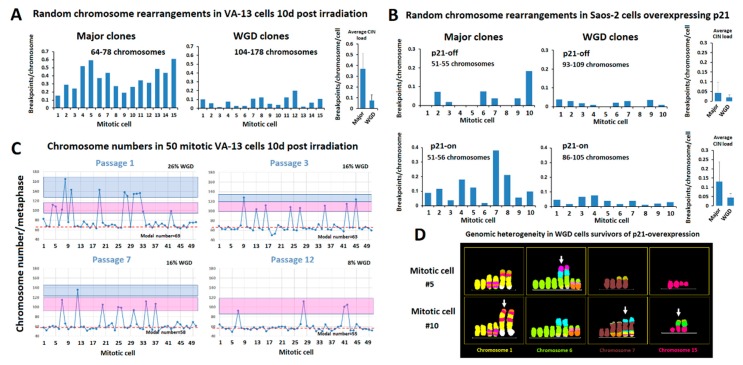
Polyploidization protects the abnormal cancer genome from ongoing structural chromosome aberrations and promotes intratumor heterogeneity. (**A**) Karyotypic analysis in two groups of 15 VA-13 cells, harvested 10 days after exposure to gamma irradiation, reveals significantly lower rates of random structural CIN in the cells that have undergone 1 or 2 rounds of WGD (composed from 104–178 chromosomes) as compared to those undergoing mitosis of the major VA-13 clones (composing of 64–78 chromosomes). Structural CIN was calculated as breakpoints/chromosome/cell. (**B**) Similar results were obtained in the osteosarcoma Saos-2 cells suffering from DNA replication stress upon prolonged p21 overexpression that duplicates the average structural CIN load. (**C**) Distribution of chromosome counts in 50 co-dividing VA-13 cells harvested in subsequent passages after 2.4 Gy of gamma irradiation (1 passage = 2–4 days in culture). Red dotted line represents chromosome numbers proximal to major clones. Pink boxes include cells that underwent one round of WGD. Blue boxes include cells that underwent more than one round of WGD. Note that despite the decline in the rates of WGD by time in culture, very few intermediate chromosome counts were recorded between the distinct ploidy indices, suggesting that cells with heavily unbalanced DNA content are less proficient at dividing. (**D**) M-FISH pseudo colored partial karyotypes of two WGD clones, generated with p21-overexpression-mediated genome reshuffling, display duplicated copies of novel complex recombinant chromosomes (arrows) not observed in non-endoreduplicated cells, suggesting that WGD increases tolerance of novel structural aberrations and thus contributes to genomic heterogeneity.

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
