# Peer review of "Karyotypic Flexibility of the Complex Cancer Genome and the Role of Polyploidization in Maintenance of Structural Integrity of Cancer Chromosomes"

_cancers, 2020, doi:10.3390/cancers12030591_

Round 1
Reviewer 1 Report
Manuscript by Raftopoulou et al showed cancer genome instability and alterations in cytogenetic levels. This manuscript contains many beautiful images and strong analysis. I am simply impressed. I believe this is one of the best manuscript I reviewed recently. I have no major comments before publications. Great research.
Author Response
We cordially thank the reviewer for the very positive response.
Reviewer 2 Report
This manuscript briefly depicts karyotypic flexibility of the complex cancer genome and the role of polyploidization in maintenance of structural integrity of cancer chromosomes. The manuscript is suitable for publication in the cancers.
Comment 1
This article is well written. It provides in-depth discussion among cancer genome and the role of polyploidization in maintenance of structural integrity of cancer chromosomes. The information obtained from cancer genome and structural integrity of cancer chromosomes will help us understand cancer. However, some of sentences are redundant, it has to be more concise and accurately reflect the points discussed.
Comment 2
It is noted that your manuscript needs careful editing for references.
Comment 3
The figure is too small. Like Fig 1 B, I can’t even watch it clearly.
The manuscript is suitable for publication in the cancers. However, it needs careful editing by author about manuscript format.
Author Response
Comment 1
This article is well written. It provides in-depth discussion among cancer genome and the role of polyploidization in maintenance of structural integrity of cancer chromosomes. The information obtained from cancer genome and structural integrity of cancer chromosomes will help us understand cancer. However, some of sentences are redundant, it has to be more concise and accurately reflect the points discussed.
We thank the reviewer for constructive comments. In the revised version we have omitted the redundant sentences from Lines: 236-238, Lines: 307-308, Lines: 315-316, and Lines: 325-327
Comment 2
It is noted that your manuscript needs careful editing for references.
All references in the revised text are double-checked. And double numbering is corrected in the revised manuscript.
Comment 3
The figure is too small. Like Fig 1 B, I can’t even watch it clearly.
In the revised version we uploaded original high-resolution Figures that are appropriate for publication. We also made some minor revisions in Figures 1 and 3 to read better.
Reviewer 3 Report
Review
Cancers-726776
“Karyotypic flexibility of the complex cancer genome and the role of polyploidization in maintenance of structural integrity of cancer chromosomes”
Raftopoulou et al.
Dear Editor,
Thank you very much for considering my expertise in the revision process of the article; “Karyotypic flexibility of the complex cancer genome and the role of polyploidization in maintenance of structural integrity of cancer chromosomes submitted for publication in the Cancers journal.
In the following article authors are trying to identify patterns of genome evolution in cancer cells driven by oncogene-replication stress, telomere dysfunction, or genotoxic therapeutic interventions. In order to achieve this, authors have examined five karyotypically-diverse clones of osteosarcoma cell line- U2-OS, by applying advance cytogenetic techniques such as mFISH/SKY molecular karyotyping, combined with inverted DAPI banding and aCGH.
Moreover, through mFISH karyotyping of a panel of human cell lines, utilizing alternative lengthening of telomeres (ALT) for telomere maintenance, characterized by highly pronounced rates of structural chromosomal instability, authors have revealed that such cells promote numerical chromosome abnormalities, which frequently results in doubling of the whole genome. According to this finding, authors speculated that numerical genome variations may protect the cancer genomes from hypermutation. As it was stated by the authors, the above hypothesis, has been further tested in ALT cells exposed to ionizing radiation, or in cells with inducible over-expression of p21, resulting in increased DNA replication-stress. With the help of cytogenetic analyses, authors revealed that although polyploidy promotes genomic heterogeneity, it also protects the complex cancer genome and hence confers genotoxic therapy resistance.
In general, the article is well presented and the experiments are adequately described and performed. The amount of work and time invested in the article seems to be substantial and this contributes significantly to the overall quality of the paper. Moreover, my opinion is that the article of such type will contribute extensively to the development of the cancer biology field. Therefore, I am very positive for publishing the current manuscript in the Cancers journal.
However, despite all superlatives related to the overall scientific and technical quality of the paper, there are some recommendations, which will be good to be taken into consideration before the final manuscript is approved for publication.
- It is quite difficult to rationalize the meaning of the graphical abstract without reading carefully the entire article. Usually the purpose of the graphical abstract is to give visual representation of the problem the article is dealing with and the findings addressing the investigated question.
- Some of the abbreviations are not indicated on their first appearance. For example, M-FISH/SKY, aCGH.
- Some of the figures are not large enough to show the details of the depicted experiments, especially those illustrating the aCGH analysis.
- Also, it will be better if the letters indicating the sub-panel in the figure are in front of the corresponding text. It is somehow confusing.
- It is good authors to justify the versatile experimental conditions, which they have used in different experimental setups. For example, sometimes 15 metaphase spreads are analyzed, sometimes 30. Also, the chromosome number in all cell lines and clones vary significantly from each other and authors have combined and analyzed most of the time cells with the most representative number of chromosomes. However, the variations sometimes are from 69-80, 115-136 or 104-178, in different experiments. It will be good if authors justified their choices and also provide a distribution of the chromosome number in different experiments.
- At some places the article is difficult to read due to some complex expressions. It is advisable authors to spend some time and reconstruct and rephrase some difficult parts in the paper to make them easily understandable.
- It is good authors to include western blot analysis validating the indicated CDT1 overexpressing clones as well as the confirmation of the Doxorubicin resistant R1 and R2 clones by clonogenic survival assay.
Author Response
We thank the reviewer for constructive comments that significantly improved the quality of our revised manuscript.
Specific comments:
1. It is quite difficult to rationalize the meaning of the graphical abstract without reading carefully the entire article. Usually the purpose of the graphical abstract is to give visual representation of the problem the article is dealing with and the findings addressing the investigated question.
We agree with the referee, hence we have restructured and enriched the information included in the revised version of our graphical abstract, that will be uploaded as a tiff for better quality
2. Some of the abbreviations are not indicated on their first appearance. For example, M-FISH/SKY, aCGH.
We incorporated appropriate indications of abbreviations as indicated in the revised text lines:113-115
3.Some of the figures are not large enough to show the details of the depicted experiments, especially those illustrating the aCGH analysis.
In the revised version we uploaded original high-resolution Figures that are appropriate for publication. We also made some minor revisions in Figures 1 and 3 to read better.
4. Also, it will be better if the letters indicating the sub-panel in the figure are in front of the corresponding text. It is somehow confusing.
The requested changes have been incorporated in all three main Figure legends.
5. It is good authors to justify the versatile experimental conditions, which they have used in different experimental setups. For example, sometimes 15 metaphase spreads are analyzed, sometimes 30.
We consider single cell analyses of 10 metaphase spreads from the same harvest/cell line, statistically enough for comparisons of random structural instability between different cell lines since hundreds of chromosomes per cell line are examined by M-FISH and Inverted-DAPI Banding (the least 500-700 chromosomes). Despite the extremely laborious process, seeking higher quality most of our structural CIN analyses were performed in 15 randomly selected spreads from the same harvest, or 15 randomly picked spreads belonging either in the major aneuploid clones or in 15 randomly selected spreads showing karyotypic evidence of endoreduplication. Criteria of selection were a) complete metaphases and b) high-quality M-FISH staining and DAPI banding pattern. In the Saos-2 cell cohort we analyzed 10 cells per condition (533-1036) chromosomes because we did not have enough high-quality mitoses of the WGD group to go to 15 cells/condition.
6. Also, the chromosome number in all cell lines and clones vary significantly from each other and authors have combined and analyzed most of the time cells with the most representative number of chromosomes. However, the variations sometimes are from 69-80, 115-136 or 104-178, in different experiments. It will be good if authors justified their choices and also provide a distribution of the chromosome number in different experiments.
Highly increased frequencies of numerical and structural CIN are characteristic of ALT cells and reflect to elevated numerical deviation between co-dividing cells. Different subclones may have different chromosome numbers for example due to the generation of a novel clonal functionally monocentric dicentric chromosome that substitutes two chromosomes with one. We showed in Sakellariou et al (2013) that clonal functionally monocentric dicentric chromosomes are frequent in human ALT cell lines. Another explanation for extensive numerical instability in the ALT-context can be provided by the frequent chromosome end-fusions that may lead to chromosome losses. In addition, a small extent of chromosome losses appears to be mitotically tolerated by the highly aneuploid major populations of every ALT cell line of our panel. These “basal” genomes are all products of an “ancestral” WGD that happened in a diploid cell and was followed by multiple chromosome losses. A higher extent of intercellular numerical instability is observed (and thus mitotically tolerated) in basal cells undergone WGD. Criteria to categorize cells in major (basal) or WGD sub-clones: a) numerical constitution b) presence of multiple copies of recombinant chromosomes. Distribution of chromosome number per every relevant experiment is indicated in Figures 1-3 and Supplementary Figures 7-15 of the revised version.
7. At some places the article is difficult to read due to some complex expressions. It is advisable authors to spend some time and reconstruct and rephrase some difficult parts in the paper to make them easily understandable.
In the new submission, we revised the graphical abstract to be more descriptive and excluded redundant sentences Lines: 236-238, Lines: 307-308, Lines: 315-316, Lines: 325-327
8. It is good authors to include western blot analysis validating the indicated CDT1 overexpressing clones as well as the confirmation of the Doxorubicin resistant R1 and R2 clones by clonogenic survival assay.
The U2-OS CDT1 TET-ON cells and the relative western blot analysis are described in Liontos et al 2007. As indicated in the revised version, before analysis these cells were grown in presence of Doxycycline for 55 days (Lines: 390-391). The R1 and R2 cells are multidrug resistant U2-OS derivatives thoroughly described in Lourda et al 2007. However, all these three cell lines U2-OS display karyotypic evidence of heavy exposure to genotoxic stressors (i.e.: multiple novel clonal chromosome rearrangements) and evidence of diverse karyotypic evolution.